# Soft Tissue Defect Reconstruction and Lymphatic Complications Prevention: The Lymphatic Flow-Through (LyFT) Concept

**DOI:** 10.3390/medicina58040509

**Published:** 2022-04-02

**Authors:** Mario F. Scaglioni, Matteo Meroni, Elmar Fritsche

**Affiliations:** Hand- and Plastic Surgery Department, Luzerner Kantonsspital, 6000 Lucerne, Switzerland; meroni369@gmail.com (M.M.); elmar.fritsche@luks.ch (E.F.)

**Keywords:** lymphatic surgery, lymphovenous anastomosis, superficial circumflex iliac artery perforator flap, deep inferior epigastric perforator flap, anterolateral thigh flap, supermicrosurgery

## Abstract

*Background and Objectives:* When a lymphatic-rich area is severely damaged, either after trauma or a surgical procedure, both soft tissue defect reconstruction and lymphatic drainage restoration are necessary. In this setting, we aim to show the potential of the lymphatic flow-through flap (LyFT) concept, which might be an attractive new solution to reduce postoperative lymphatic complications. *Materials and Methods:* Between 2018 and 2021, 12 patients presenting a soft tissue defect involving damage to the lymphatic drainage pathway received a lymphatic flow-through flap for volume and lymphatic drainage restoration. Different flaps were employed: 3 pedicled superficial circumflex iliac artery perforator (SCIP) flaps, 2 free SCIP flaps, 3 pedicled deep inferior epigastric perforator (DIEP) flaps, 2 pedicled vertical posteromedial thigh (vPMT) flaps, and 2 pedicled anterolateral thigh (ALT) flaps. A range of 1 to 3 lymphovenous anastomosis (LVA) with flap’s veins was performed (mean 1.9). For a better dead space obliteration, an additional vastus lateralis muscle flap was performed in one case. Indocyanine green (ICG) lymphography was used in all cases to identify the lymphatic pathway, make the preoperative markings, and check the patency of the anastomoses. *Results:* In all cases, the reconstructive results were satisfactory from both the functional and aesthetic points of view. No secondary surgeries were required, and only one minor complication was encountered: an infected seroma that was managed conservatively. The mean follow-up was 9.9 months (range 6–14 months). *Conclusions:* Lymphatic flow-through flaps seem to effectively reduce the risk of lymphatic complications after the reconstruction of soft tissue defects with a compromised lymph pathway. This is a versatile solution that might be used in different body regions resorting to different flap types.

## 1. Introduction

The continuous progress of microsurgical techniques has allowed surgeons to reconstruct an extremely wide range of defects throughout the body. Expectations have also grown, and simple coverage is no more considered a completely satisfactory result. Nowadays, the goal of a plastic surgeon is to restore good physical and physiological function. This is a particularly critical issue when lymphatic vessels are severely damaged. Impairment of lymph drainage often leads to a series of complications, ranging from edema to severe cellulitis, which may cause very debilitating conditions [1,2,3]. For this reason, volume restoration alone is often not sufficient since it is also necessary to prevent lymphatic sequelae [4].

Regarding dead space obliteration, many alternatives are available. In order to reduce donor site morbidity and quicken the harvest, nowadays, perforator-based flaps are preferred, either in pedicled or free form [5]. Among these, the most employed are the anterolateral thigh (ALT) flap [6], the superficial circumflex iliac artery perforator (SCIP) flap [7], and the deep inferior epigastric perforator (DIEP) flap [8], each of them with its own features.

Different procedures aimed at the restoration of lymph drainage have been described so far, but all of them are still debated, with no clear advantages of one over the others [9]. However, lymphovenous anastomosis (LVA) is gaining consistent approval to treat lymphocele and lymphedema throughout the body. It consists of shunting the lymphatic flow into venous circulation before the impaired area, providing an alternative drainage route [10].

The lymphatic flow-through (LyFT) flap is an interesting and modern concept that tries to combine both of these treatments. It not only provides healthy tissue for defect reconstruction but also allows the exploitation of the collateral veins of the flap for the LVA. This approach is particularly helpful when no suitable vessels can be found near the defect, such as after radical debulking procedures combined with radiotherapy or after severe trauma [11].

In the present article, we present our experience of a 12-patient series successfully treated with LyFT flaps for the reconstruction of defects in different body regions and with various etiologies.

## 2. Materials and Methods

Twelve patients presenting a soft tissue defect involving damage to the lymphatic drainage pathway were included in this retrospective report; they received a lymphatic flow-through flap for volume and lymphatic drainage restoration (Table 1); 6 were females, and 6 were males (50:50 gender ratio). The median age was 57 years old (range 42–82); 8 patients presented no comorbidities, 2 were affected by chronic arterial hypertension, and 2 by diabetes mellitus.

The cause of the defect was surgical tumor excision in 10 cases (8 because of sarcoma and 2 because of squamous cell carcinoma), while in 2 cases, the defect was due to trauma. The defect was localized as follows: 3 in the groin region, 2 in the abdomen and groin, 1 in the groin and medial thigh, 2 in the medial thigh, 2 in the upper thigh, 1 in the lower leg, and 1 in the upper extremity. Different types of flaps were employed, either pedicled or free. In 3 cases, we resorted to a pedicled superficial circumflex iliac artery perforator (SCIP) flap, in 2 to a free SCIP flap, in 3 to a pedicled deep inferior epigastric perforator (DIEP) flap, in 2 to a pedicled vertical posteromedial thigh (vPMT) flap, and in 2 to an anterolateral thigh (ALT) flap. The number of lymphovenous anastomoses performed with flap’s veins ranged between 1 and 3 (mean 1.9). In 7 cases, we employed a superficial flap vein, in 4 a pedicle vein, and in 1 case, we used the deep branch of the pedicle vein. Indocyanine green (ICG) lymphography was always performed preoperatively to visualize the lymphatic pathway and intraoperatively to identify lymphatic leakages and to confirm the patency of LVAs. Lymphoscintigraphy was routinely performed 6 months after surgery, confirming a sufficient lymphatic flow.

### Surgical Technique

Preoperative indocyanine green (ICG) lymphography was always performed in order to visualize and draw the pathway of the lymphatic vessels nearby the affected area. After the debulking or explorative surgery, an additional ICG scan was made to identify the main distal leaking vessels, which were then carefully isolated and prepared for anastomosis. During the flap harvest, particular care was required in order to also isolate one or more superficial reflux-free veins suitable for the LVAs. The location of these veins is also important: at this stage, the surgeon should know how the flap will be managed and inset since these veins must match the site of the previously identified leaking lymphatics. In our experience, we always performed single LVAs in an end-to-end fashion with nylon 12-0 stitches (Figure 1). ICG imaging was always performed after the anastomoses to confirm their function (Figure 2).

## 3. Results

In all cases, the reconstructive results were satisfactory from both the functional and aesthetic points of view, with full volume and range of motion restoration. The total duration of surgery ranged from 3:50 to 6:10 h. The mean follow-up period was 9.9 months (ranging from 6 to 14 months). During this period, 11 patients showed no complications, while 1 patient developed an infected seroma, which was conservatively treated with percutaneous drainage and antibiotics. No signs of lymphocele nor lymphedema were observed in any of the cases. Lymphoscintigraphy was routinely performed 6 months after surgery, confirming a sufficient lymphatic flow. In all cases, no secondary procedures were required.

### Case Report

A 67-year-old man presented an extremely large abdominal mass, which was diagnosed as soft tissue sarcoma after an open surgical biopsy. The gastrointestinal surgeons removed the whole tumor, exposing the entire bowel, and reconstructed the abdominal region bowel with a mesh (Figure 3). A 25 cm × 18 cm defect remained in the inguinal area, and a pedicled ALT flap was planned to fill the defect. In the inguinal defect, both superficial and deep lymphatics were identified and isolated with intraoperative ICG lymphography. During the elevation of the flap, a long vein originating from the pedicle was harvested and prepared for anastomosis (Figure 4). Using 3 branches of this pedicle vein, 3 LVAs were then performed, 2 with the superficial lymphatic vessels and 1 with the deep one. Their patency was proven using ICG lymphography (Figure 5). At 6 months follow-up, the reconstructive result was good. The soft tissue coverage was stable without tumor recurrence, and the lymphoscintigraphy did not show any sign of lymph stasis (Figure 6).

## 4. Discussion

Tissue defects reconstruction throughout the body represents one of the most common tasks for plastic surgeons. Either pedicled or free flaps are well known as a versatile armamentarium, and they represent the most common treatment in these cases. Different types of flaps have been prosed over the years; however, the actual trend is to prefer the perforator-based ones since they reduce donor site morbidity and allow a quicker dissection [12,13]. Among these, the typical ones are the deep inferior epigastric perforator (DIEP) flap, the anterolateral thigh (ALT) flap, and the superficial circumflex iliac artery perforator (SCIP) flap. When a sufficient amount of abdominal fat is present, the DIEP flap is considered the gold standard for autologous breast reconstruction [14], and it is widely used for soft tissue defects in many other districts [8]. The ALT flap is one of the main alternatives; it has a long pedicle and low donor site morbidity and offers the advantage of chimeric forms, including the vastus lateralis muscle [6]. Lately, the SCIP flap has been gaining approval because of its very low donor site morbidity, versatility, and aesthetic result. It also allows chimeric transfer, including muscle, nerve, and bone [15].

When the defects are large and compromise the lymphatic drainage network, the prevention of postoperative complications is of crucial importance. Chronic lymphorrhea, lymphocele, and lymphedema might develop, leading to severe discomfort for the patient, with heaviness sensation in the limbs, swelling, pain, erythema, recurrent cellulitis, and even range-of-motion limitations [16]. The best approach for this situation is to prevent lymph stasis immediately after surgery.

Different techniques have been proposed to treat lymphatic sequelae, but there is still a lack of consensus concerning their efficacy. The most validated options nowadays are lymphovenous anastomosis (LVA) [10], vascularized lymphnode transfer (VLNT) [17], and lymphatic tissue transfer [18]. LVA consists of diverting the lymph flow into the venous circulation upstream of the damage, offering the lymph an alternative draining route. This is performed by means of microsurgical or supermicrosurgical (when the vessel’s diameter is <0.8 mm) anastomosis between one or more functioning lymphatic vessels with a nearby reflux-free vein. This is an essential point since, to obtain optimal lymph drainage, it is strictly necessary to prevent backflow. In this way, even the low pressure coming from the lymphatic system is sufficient to overcome the resistance and reach the circulation [19]. Moreover, this procedure has been recently described as a potentially effective treatment for the management of long-standing ulcers associated with chronic venous insufficiency and lymphorrhea [20]. VLNT, instead, involves the transfer of functional lymph nodes, with microanastomosis with vasculature in the recipient bed to maintain their blood supply, to restore the physiological lymphatic flow. However, the underlying mechanism of action of this procedure remains unclear [21,22]. Lymphatic tissue transfer is another fascinating option that relies on neolymphangiogenesis between donor and recipient vessels, stimulated by the transfer of healthy tissue [23,24,25]. Compared to VLNT, this procedure offers the advantage that it does not impair the lymphatics of an otherwise healthy area and exploits the same tissue used for defect reconstruction.

The lymphatic flow-through (LyFT) concept combines the soft tissue transfer for volume restoration with the possibility of reducing the risk of lymphatics sequelae. It employs healthy veins from the transferred flap to perform one or more LVAs in the affected area. Moreover, when we resort to SCIP or DIEP, it also allows us to move lymphatic-rich tissue that may stimulate neolymphangiogenesis, as previously described. Flow-through flaps have already been presented in the literature by Fujiki M. et al. [26] to reconstruct arterial and venous defects, and then by di Summa P. and Guiller D. [11] to restore lymphatic flow in inguinal defects. This solution is particularly attractive in all those cases where no suitable nearby veins can be found, either because of severe traumas or iatrogenic damage (such as after extensive debulking procedures and radiotherapy).

In our case series, we used both the superficial veins of the flap and the veins of the pedicle. In particular, resorting to the SCIP flap, we have the opportunity to choose either the deep or the superficial branch of the superficial circumflex iliac vein. We have already resorted to a similar approach, employing the deep branch vein as a donor vessel for LVA to prevent donor site lymphocele after SCIP flap harvest [27]. In the DIEP, we have many superficial veins available, but we could also use branches coming from the pedicle. Similarly, the ALT flap has a long pedicle presenting suitable venous branches.

From the technical point of view, it is essential to mention the role of intraoperative ICG lymphography. This tool is of paramount importance to identify the leaking interrupted lymphatic vessels and, hence, to shunt them into the venous circulation [28]. A feasible alternative would be closing these vessels, blocking the leakage and preventing lymphocele development, but it would imply a much higher risk of lymphedema. Another very important examination is lymphoscintigraphy. We routinely rely on it preoperatively to map the lymphatic network in the affected area and during the follow-up to confirm the efficacy of the procedure. This is the most accurate method to visualize and assess lymph flow restoration after the reconstruction [29,30]. A minor limitation of this procedure can be the additional surgical time required to isolate the recipient vessels for the LVAs and to execute the anastomoses. These procedures require about 40 to 60 min when performed by an experienced surgeon

The number of cases described in the literature dealing with this technique is still low; however, we believe that this is a very promising procedure that is worthy of further study. Our results seem to confirm its efficacy, and we suggest taking it into account for the reconstruction of soft tissue defects where the lymphatic network is widely compromised.

The most significant limitation of this report is the limited number of cases with heterogenous defects and patient characteristics. This compromises the possibility of gaining statistically significant results. However, as previously mentioned, this is a very modern surgical approach for particularly complex defects. In this setting, it is extremely difficult to gather a large cohort of patients, and further studies are necessary to confirm the efficacy of this procedure.

## 5. Conclusions

This report is intended to show, taking into account the aforementioned limitations, the potential of a modern, multi-effective approach to the reconstruction of large soft tissue defects with significant lymphatic impairment. The lymphatic flow-through concept may allow us to fully exploit the potential of either free or pedicled tissue transfer, combining good coverage with the immediate restoration of the lymphatic drainage in order to prevent immediate and long-term lymphatic complications.

## Figures and Tables

**Figure 1 medicina-58-00509-f001:**
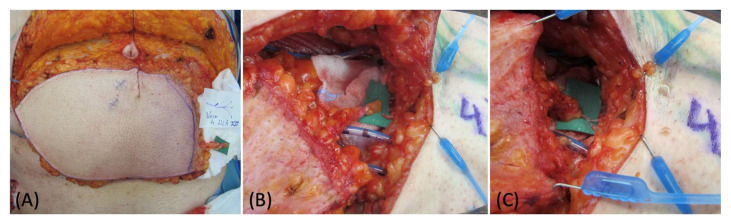
(**A**) DIEP flap harvest with lateral superficial vein isolation. (**B**) Flap preparation at recipient site before and after the LVA (**C**).

**Figure 2 medicina-58-00509-f002:**
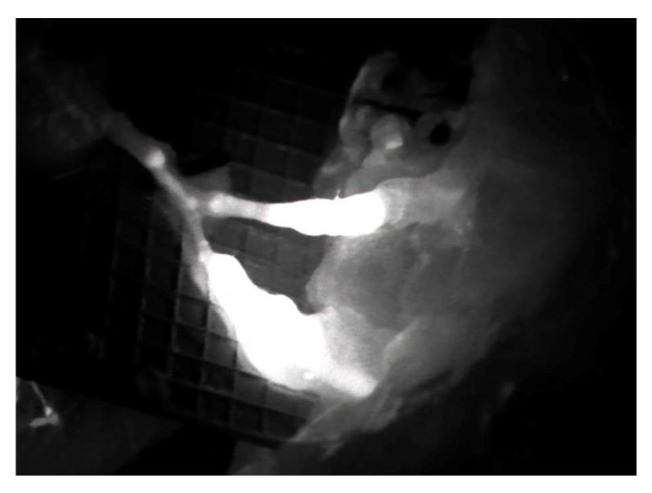
Intraoperative ICG imaging to check and confirm the patency and function of the LVA.

**Figure 3 medicina-58-00509-f003:**
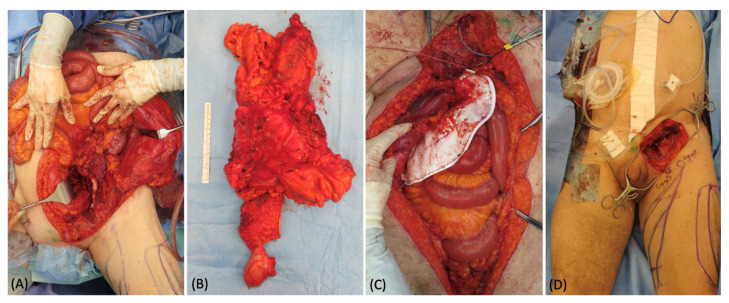
(**A**) Intraoperative picture of the extensive surgery for the sarcoma removal in the abdomen. (**B**) 45 × 30 cm specimen. (**C**) Abdomen reconstruction with a mesh. (**D**) Appearance of the groin defect at the end of the abdominal surgery.

**Figure 4 medicina-58-00509-f004:**
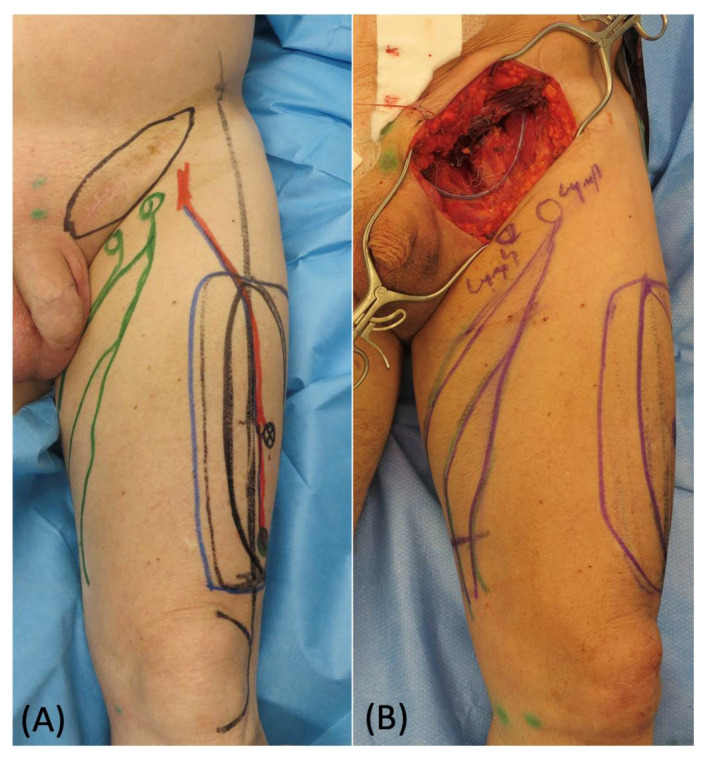
(**A**) Preoperative picture of skin marking of the ALT flap with ICG-guided lymphatic ducts detection. (**B**) Remaining groin defect after sarcoma removal.

**Figure 5 medicina-58-00509-f005:**
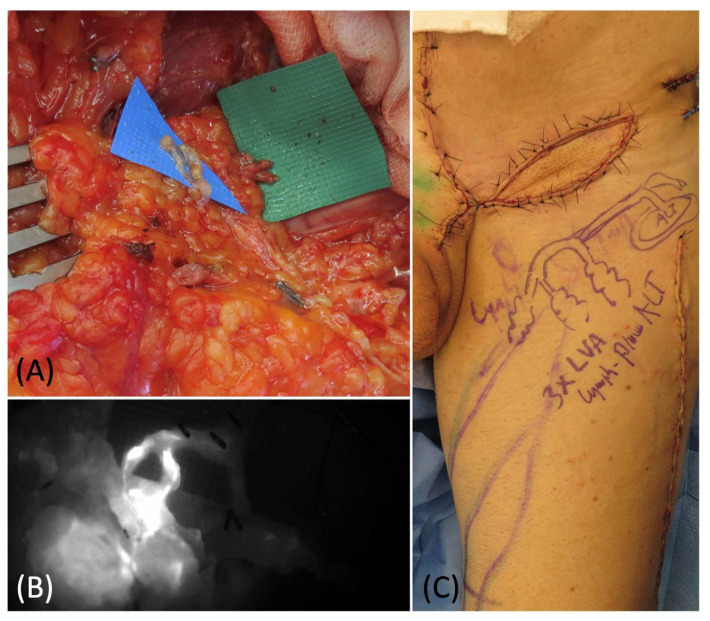
(**A**) Lymphovenous anastomoses of 2 superficial lymphatics and 1 deep lymphatic with 3 branches of the pedicle vein. (**B**) Intraoperative ICG lymphography to confirm the patency of the anastomoses. (**C**) Picture of the groin at the end of the procedure: the ALT flap was inset and 3 LVAs were performed with 3 pedicles branches.

**Figure 6 medicina-58-00509-f006:**
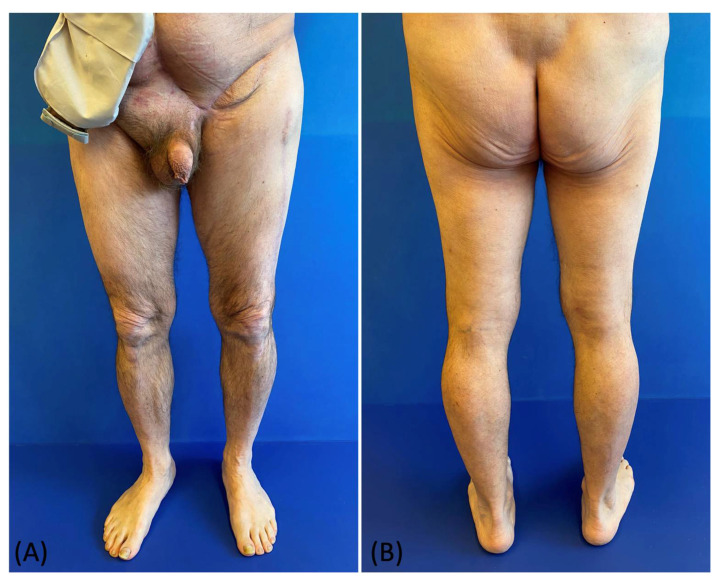
(**A**,**B**) Postoperative picture at 6 months follow-up: frontal view and posterior view.

**Table 1 medicina-58-00509-t001:** Patient demographic and case characteristics.

Patient	Gender	Age	Etiology	Location	Comorbidities	Flap	Recipient Vein for LVA	Number LVA	Complications	Follow-Up (Months)	Functional Outcomes/Aesthetic Result
1	M	63	Trauma	Medial thigh	DM	Pedicled DIEP	Superficial Flap Vein	3	None	14	Full ROM/++
2	F	56	Trauma	Lower leg	HTN	Free SCIP	Deep Branch Pedicle Vein	2	None	14	Full ROM/++
3	F	76	Sarcoma	Groin	None	Pedicled DIEP	Superficial Flap Vein	3	Infected Seroma	12	Full ROM/+
4	F	65	Sarcoma	Upper extremity	None	Free SCIP	Superficial Flap Vein	1	None	11	Full ROM/+
5	M	67	Sarcoma	Intra-abdominal/Groin	None	Pedicled ALT	Pedicle Vein	3	None	11	Full ROM/++
6	M	82	Sarcoma	Intra-abdominal/Groin	HTN	Pedicled ALT + VLM	Pedicle Vein	3	None	11	Full ROM/+
7	F	42	Sarcoma	Groin/Medial thigh	None	Pedicled SCIP	Superficial Flap Vein	1	None	11	Full ROM/+
8	M	45	Skin Tumor	Medial thigh	None	Pedicled SCIP	Superficial Flap Vein	2	None	11	Full ROM/++
9	M	47	Skin Tumor	Groin	None	Pedicled vPMT	Pedicle Vein	1	None	6	Full ROM/+
10	M	59	Sarcoma	Upper thigh	DM	Pedicled DIEP	Superficial Flap Vein	2	None	6	Full ROM/+
11	F	39	Sarcoma	Upper thigh	None	Pedicled SCIP	Superficial Flap Vein	1	None	6	Full ROM/+
12	F	51	Sarcoma	Groin	None	Pedicled vPMT	Pedicle Vein	1	None	6	Full ROM/+

DM: diabetes mellitus; HTN: hypertension; SCIP: superficial circumflex iliac artery perforator; DIEP: deep inferior epigastric artery perforator; ALT: anterolateral thigh flap; VLM: vastus lateralis muscle; vPMT: vertical posteromedial thigh; LVA: lymph venous anastomosis; ROM: range of motion; +: good aesthetic result; ++: very good aesthetic result.

## Data Availability

Data are available from the authors upon reasonable request.

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
