# Peer review of "Soft Tissue Defect Reconstruction and Lymphatic Complications Prevention: The Lymphatic Flow-Through (LyFT) Concept"

_medicina, 2022, doi:10.3390/medicina58040509_

Round 1

Reviewer 1 Report

The authors conducted a retrospective outcome analysis of twelve patients who have received a lymphatic flow-through flap for volume and lymphatic drainage reconstruction. The manuscript is well written and organized. However, the study group is very heterogenous and not large enough in order to gain statistical and clinical significance. A much larger cohort and a control group would be necessary. Furthermore, study limitations should be discussed more.

Author Response

Reviewer 1

The authors conducted a retrospective outcome analysis of twelve patients who have received a lymphatic flow-through flap for volume and lymphatic drainage reconstruction. The manuscript is well written and organized. However, the study group is very heterogenous and not large enough in order to gain statistical and clinical significance. A much larger cohort and a control group would be necessary. Furthermore, study limitations should be discussed more.

- Answer

Dear Reviewer, thank you for your appreciation and comments. We agree that the number of patients is limited, however, this it is almost inevitable when dealing with such a peculiar type of treatment. We believe that more studies from different authors will be necessary in order to support this concept with a statistical and clinical significance. For this reason, we also believe that is important to share similar experience in order to make this concept known and let it be replicated.

We now modified the Discussion section of the manuscript pointing out this issue.

Reviewer 2 Report

Dear authors, this is an interesting report. The clinical cases presented are relevant and results are very good. 

The word "study" should be replaced with the word " report" because no statistical nalysis was performed

Introduction

Line 55 “In the present article, we present our experience of an eight patient’ series success-55 fully treated with LyFT flaps”. Throughout  the manuscript 12 patients were reported. Please clarify.

Materials and methods

Syntax should be revised to improve readability

Surgical technique section should be written in past tense in the paragraphs where it is deemed necessary

Results

Line 108. “Lymphoscintigraphy has been routinely performed 6 months after surgery, confirming a sufficient lymphatic flow.” This data should be provided also in materials and methods section

Caption of figure 5 reported a figure D; however, figure D was not provided.

Discussion

Line 165-167 “…anastomosis between one or more functioning lymphatic vessels with a nearby reflux-free vein. This is an essential point since to obtain optimal lymph drainage, it is strictly necessary to prevent backflow”. A reflux-free vein is the best condition; however, LVA was proved to be an effective strategy also in case of chronic venous insufficiency, and should be discussed.

Cigna E et al. Lymphatico-venous anastomosis in chronic ulcer with venous insufficiency: A case report. Microsurgery. 2021 Sep;41(6):574-578. doi: 10.1002/micr.30753

Conclusions section should be added to the manuscript

Author Response

Reviewer 2

Dear authors, this is an interesting report. The clinical cases presented are relevant and results are very good. 

The word "study" should be replaced with the word " report" because no statistical nalysis was performed

Introduction

Line 55 “In the present article, we present our experience of an eight patient’ series success-55 fully treated with LyFT flaps”. Throughout the manuscript 12 patients were reported. Please clarify.

Materials and methods

Syntax should be revised to improve readability

Surgical technique section should be written in past tense in the paragraphs where it is deemed necessary

Results

Line 108. “Lymphoscintigraphy has been routinely performed 6 months after surgery, confirming a sufficient lymphatic flow.” This data should be provided also in materials and methods section

Caption of figure 5 reported a figure D; however, figure D was not provided.

Discussion

Line 165-167 “…anastomosis between one or more functioning lymphatic vessels with a nearby reflux-free vein. This is an essential point since to obtain optimal lymph drainage, it is strictly necessary to prevent backflow”. A reflux-free vein is the best condition; however, LVA was proved to be an effective strategy also in case of chronic venous insufficiency, and should be discussed.

Cigna E et al. Lymphatico-venous anastomosis in chronic ulcer with venous insufficiency: A case report. Microsurgery. 2021 Sep;41(6):574-578. doi: 10.1002/micr.30753

Conclusions section should be added to the manuscript

- Answer

Dear Reviewer, thank you for your appreciation and comments. We now answer them point to point as follows:

Introduction

Line 55 “In the present article, we present our experience of an eight patient’ series success-55 fully treated with LyFT flaps”. Throughout the manuscript 12 patients were reported. Please clarify.

  • Twelve patients were included in this study as shown in Table 1. We corrected this error.

Materials and methods

Syntax should be revised to improve readability

Surgical technique section should be written in past tense in the paragraphs where it is deemed necessary

  • The manuscript has been submitted to English-native reviewer for language revision.
  • The tense in the surgical technique section has been corrected to past tense.

Results

Line 108. “Lymphoscintigraphy has been routinely performed 6 months after surgery, confirming a sufficient lymphatic flow.” This data should be provided also in materials and methods section

Caption of figure 5 reported a figure D; however, figure D was not provided.

  • We added that data also in the materials and methods section.
  • We corrected that typo in Figure 5 legend, there is no Figure D.

Discussion

Line 165-167 “…anastomosis between one or more functioning lymphatic vessels with a nearby reflux-free vein. This is an essential point since to obtain optimal lymph drainage, it is strictly necessary to prevent backflow”. A reflux-free vein is the best condition; however, LVA was proved to be an effective strategy also in case of chronic venous insufficiency, and should be discussed.

Cigna E et al. Lymphatico-venous anastomosis in chronic ulcer with venous insufficiency: A case report. Microsurgery. 2021 Sep;41(6):574-578. doi: 10.1002/micr.30753

Conclusions section should be added to the manuscript

  • We added a short mention to that interesting report in the discussion.
  • A conclusion section has been added.

Reviewer 3 Report

Dear Editor and Authors,

Thank you for the opportunity to review the manuscript entitled “Soft tissue defect reconstruction and lymphatic complications prevention: the Lymphatic Flow-Through (LyFT) concept.” The authors presented twelve patients with a soft tissue defect involving a damage at the lymphatic drainage pathway who received a lymphatic flow-through flap for volume and lymphatic drainage restoration. The authors concluded that lymphatic flow-through flaps seem to effectively reduce the risk of lymphatic complications after the reconstruction of soft tissue defects with a compromised lymph pathway. They suggest it is a versatile solution that might be used in different body regions resorting to different flap types.  The study presents innovative concept and should be of interest of most reconstructive surgeons.
I have some additional suggestions

  • Abstract – please: add the aim of the study, explain abbreviations when used for the first time, key words should also not include abbreviations
  • Provide a clear aim of the study
  • Were there eight /line 55/ or twelve patients /line 59/?
  • Table 1 footnote – explain “+/++” for aesthetic results evaluation
  • Results – could you provide length of the procedures? And further discuss a possible limitation of the technique which may be lengthening of the procedure (how long?).
  • To highlight the importance of the presented technique you could provide some statistical data on lymphatic sequelae after such reconstructions

Congratulations on using such advanced techniques in reconstructions!

Author Response

Review 3

Thank you for the opportunity to review the manuscript entitled “Soft tissue defect reconstruction and lymphatic complications prevention: the Lymphatic Flow-Through (LyFT) concept.” The authors presented twelve patients with a soft tissue defect involving a damage at the lymphatic drainage pathway who received a lymphatic flow-through flap for volume and lymphatic drainage restoration. The authors concluded that lymphatic flow-through flaps seem to effectively reduce the risk of lymphatic complications after the reconstruction of soft tissue defects with a compromised lymph pathway. They suggest it is a versatile solution that might be used in different body regions resorting to different flap types.  The study presents innovative concept and should be of interest of most reconstructive surgeons.
I have some additional suggestions

  • Abstract – please: add the aim of the study, explain abbreviations when used for the first time, key words should also not include abbreviations
  • Provide a clear aim of the study
  • Were there eight /line 55/ or twelve patients /line 59/?
  • Table 1 footnote – explain “+/++” for aesthetic results evaluation
  • Results – could you provide length of the procedures? And further discuss a possible limitation of the technique which may be lengthening of the procedure (how long?).
  • To highlight the importance of the presented technique you could provide some statistical data on lymphatic sequelae after such reconstructions

Congratulations on using such advanced techniques in reconstructions!

- Answer

Dear Reviewer, thank you for your appreciation and comments. We now answer them point to point as follows:

  • Abstract – we corrected those issues
  • We clarified the aim of the study
  • Twelve patients were included in this study as shown in Table 1. We corrected this error.
  • We clarified the +/++ meaning
  • We now provide the indicative range of time required for the procedures included in this work. We also. Mentioned the additional time required to perform the LVAs as a minor limitation, as you rightly suggested.
  • Unfortunately, the number of patients is still too limited to obtain statistically relevant results. This it is almost inevitable since we are dealing with a very peculiar type of treatment. We believe that more studies from different authors will be necessary in order to support this concept with a statistical and clinical significance. For this reason, we also believe that is very important to share similar experience in order to make this concept known and validated.

Round 2

Reviewer 1 Report

The authors adressed all reviewer comments in a well manner and I think, this work might be of great interest for the jounnals readership!

Author Response

Thank you

Correct

Reviewer 2 Report

A minor thing. Regarding case report (page 3), ALT flap should mentioned  also in the caption of figure 4 and 5

Author Response

Thank you 

correct